# Invariant Learning with Annotation-free Environments

**Phuong Quynh Le**[1] **Christin Seifert**[1] **Jörg Schlötterer**[1,2]

[1]**University of Marburg,** [2]**University of Mannheim**
{phuong.le,christin.seifert,joerg.schloetterer}@uni-marburg.de

## Abstract

Invariant learning is a promising approach to improve domain generalization compared to Empirical Risk Minimization (ERM). However, most invariant learning methods rely on the assumption that training examples are pre-partitioned into different known environments. We instead infer environments without the need for additional annotations, motivated by observations of the properties within the representation space of a trained ERM model. We show the preliminary effectiveness of our approach on the ColoredMNIST benchmark, achieving performance comparable to methods requiring explicit environment labels and on par with an annotation-free method that poses strong restrictions on the ERM reference model.

## 1 Introduction

Empirical Risk Minimization (ERM) is known to generalize poorly when the test data has a different distribution than the training data. When features have a statistical but not a causal relationship with class labels, we call this a *spurious correlation* and the corresponding features *spurious features*. For example, the Waterbirds dataset [Sagawa *et al.*, 2020] has background as a spurious feature. During training, 95% of waterbirds have water background and 95% of landbirds have land background, whereas land and water background are equally distributed in the test set, causing a distribution shift. This shift results in poor predictive accuracy of ERM models, particularly driven by the inferior performance of groups that lack the spurious correlation (e.g., waterbirds on land). Focusing on individual groups (formed by combinations of class labels and spurious features), Sagawa *et al.* [2020] and Srivastava *et al.* [2020] proposed distributionally robust optimization methods to improve the worst group accuracy. Generalizing the setting, methods aim to learn invariant features across different so-called environments. These environments are partitionings of the data into subsets with varying presence of the spurious correlation (e.g., only 80% of waterbirds with water background in one environment), simulating data collection under varying circumstances. Approaches that focus on finding invariant representations of data across different training environments encompass optimization by a gradient penalty on classifier weights [Arjovsky *et al.*, 2019] or by min-max frameworks [Krueger *et al.*, 2021]. DecAug [Bai *et al.*, 2021] learns disentangled features that separately capture category and context information and enhances generalization through a data augmentation method.

However, these approaches have limitations in realistic scenarios. Consider for example the prediction of skin cancer (benign vs. malignant lesions) in the ISIC dataset [Codella *et al.*, 2019]. In this dataset, almost half of the benign cases contain a colored patch that had been applied to the patient's skin in the hospital. None of the malignant cases contain a patch. Thus, half of the benign cases can be easily identified by the colored patch alone [Nauta *et al.*, 2021]. In general, clinical markers, such as ruler measurements, surgical marks, or other artifacts, tend to become spurious features and reduce the generalization ability of ERM models [Mishra and Celebi, 2016; Pewton and Yap,

2022]. To generate training environments for invariant learning methods, it would be necessary to (i) identify all potential spurious features and (ii) artificially generate samples for the missing groups, i.e., groups with spurious features and uncorrelated class labels. Since not all artifacts may be known in advance and human annotation, especially by domain experts, is costly, the requirement to create suitable environments is unrealistic in such settings. EIIL [Creager *et al.*, 2021] eliminates this requirement and instead infers appropriate environments from a reference model. Specifically, EIIL seeks to construct environments with maximal (anti)correlations, i.e., a feature is highly correlated with one class in one environment and another class in another environment, thereby forcing the prediction model to rely only on invariant features (causal for the prediction). However, this approach depends on a reference model that relies heavily on spurious correlations and requires estimating the distribution of assignments of instances to environments.

Inspired by the observation by Le *et al.* [2024] that ERM induces clusters of samples with the same spurious features in the representation space and that these clusters can be exploited to find samples where the spurious correlation is missing, we adopt their strategy to infer suitable environments for invariant learning. We show that the adoption succeeds in finding counter-spurious correlation samples (*conflict samples*) from which we can construct environments that allow to learn invariant features. Our approach does not pose any requirements on the reference model and accordingly leaves the training process untouched. Clustering on the representation space of a trained model introduces little overhead, which makes our method an efficient approach to infer environments for invariant learning without the need for annotations.

Our code is available at: `https://github.com/aix-group/prusc`.

## 2  Background on Invariant Risk Minimization

Empirical Risk Minimization (ERM) [Vapnik, 1991] minimizes the error across training data, using all features

$$\min_{f:\mathcal{X}\to\mathcal{Y}} \frac{1}{n} \sum_{i=1}^{n} L(f(x_i), y_i), \tag{ERM}$$

where $L$ is the loss function, $f$ maps from input space $\mathcal{X}$ to the output space $\mathcal{Y}$, $f(x_i)$ is the predicted value and $y_i$ is the true value.

Invariant Risk Minimization [Arjovsky *et al.*, 2019] aims to extract environment-invariant features from input data to enable consistent predictions across environments.

$$\min_{\Phi:\mathcal{X}\to\hat{\mathcal{H}},\omega:\hat{\mathcal{H}}\to\mathcal{Y}} \sum_{e\in E_{train}} R^e(\omega \circ \Phi) \text{ subject to } \omega \in \arg\min_{\tilde{\omega}:\hat{\mathcal{H}}\to\hat{y}} R^e(\tilde{\omega} \circ \Phi), \forall e \in E_{train}, \tag{IRM}$$

$\hat{\mathcal{H}}$ is the invariant feature space, $\mathcal{Y}$ is the output space, $R^e(\Phi)$ is the risk under a known environment $e \in E_{\text{train}}$. IRM learns the function $f = \omega \circ \Phi$ where $\Phi$ learns the invariant features from multiple environments. The final prediction is made by $\omega$ based on the extracted invariant feature space $\hat{\mathcal{H}}$. For practical reasons, Arjovsky *et al.* [2019] simplify the dual-objective optimization to

$$\min_{\Phi:\mathcal{X}\to\mathcal{Y}} \sum_{e\in\mathcal{E}_{\text{tr}}} R^e(\Phi) + \lambda \cdot \left\| \nabla_w \big|_{w=1.0} R^e(w \cdot \Phi) \right\|^2 \tag{IRMv1}$$

## 3  Dataset

We use ColoredMNIST [Arjovsky *et al.*, 2019], a synthetic dataset constructed from MNIST where colors are strongly correlated with the class labels. The dataset is constructed as follows:

1. Set label $\tilde{y} = 1$ for images with digits from 0 to 4, otherwise $\tilde{y} = 0$.
2. The final label $y$ is defined by randomly flipping 25% of the labels $\tilde{y}$ (noise level $n_y = 0.25$).
3. The color id $z$ is defined by flipping $y$ with probability $p_e$ and images are colored red if $z = 0$ and green if $z = 1$ (see Fig. 1a for examples)

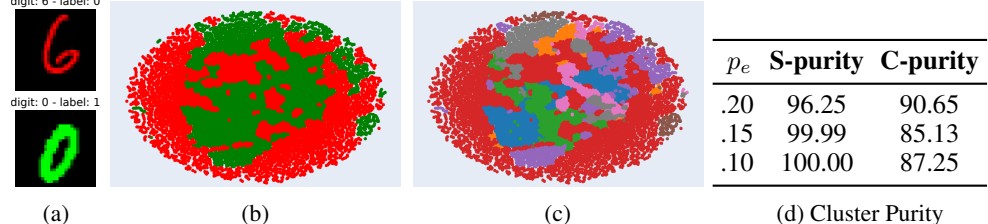

| | | | $p_e$ | **S-purity** | **C-purity** |
|---|---|---|---|---|---|
| | | | .20 | 96.25 | 90.65 |
| | | | .15 | 99.99 | 85.13 |
| | | | .10 | 100.00 | 87.25 |
| (a) | (b) | (c) | | (d) Cluster Purity | |

Figure 1: (a) Example instances from ColoredMNIST with colors correlated with binary class labels. t-SNE [Van der Maaten and Hinton, 2008] projected embedding space with colors representing (b) color annotations and (c) clusters obtained by k-means clustering. (d) Cluster purity w.r.t. spurious features (S-purity) and classes (C-purity).

For the training set, the noise level $n_y = 0.25$ and the color correlation level is $p_e = 0.15$.[1] The standard test set has the same noise level ($n_y = 0.25$), but an inverse color correlation $p_e = 0.9$. In Sec. 6 we analyze performance for different color correlation levels, i.e., test sets with varying $p_e$.

## 4 Method

We first present initial observations on the ERM representation space trained in a spurious setting, and subsequently derive our annotation-free sampling to construct environments for IRM training. A key contribution of our method is the identification of conflict samples, i.e., instances with a relation between spurious attributes and class labels that is opposite to the spurious correlation present in the training data. Our method does not pose any restrictions on the model or training process, but solely relies on properties of the learned representation space.

### 4.1 Initial Observations

We train an ERM model on ColoredMNIST and extract the embedding of the penultimate layer. We perform $k-$means clustering[2] on the embedding of the training set and obtain the clusters as visualized in Fig. 1c. Interestingly, the purity of the clusters is higher w.r.t. spurious features than w.r.t. class labels (99.99 vs. 85.13, cf. Tab. 1d, $p_e = 0.15$). Therefore, we assume that each cluster defines a spurious feature. Since spurious features have strong correlations with a particular class, a cluster's purity is usually high with respect to both, the spurious feature and the class label. For our binary classification task, we define *minority cases* as samples from the class with fewer samples within a cluster. These samples share similar spurious features but have class labels that are not correlated with these features. In other words, *minority cases* in a cluster *conflict* with the spurious correlations introduced in the training set (in short, *conflict* samples). We base our environment construction on this observation.

**Discussion.** While our initial analysis in this paper focuses on the ColoredMNIST dataset, which is simple and has strong spurious correlations (with a spurious ratio of 0.9 in the training data), related studies have shown that spurious features are typically learned early in training or are easier for models to learn, leading models to rely on these features [Shah *et al.*, 2020]. Therefore, we hypothesize that the observation that representations cluster stronger w.r.t. to spurious features than w.r.t. class labels also holds in scenarios with more realistic or complex spurious correlations (e.g., multiple spurious correlations as in the CelebA dataset [Liu *et al.*, 2015]).

### 4.2 Annotation-free Environment Construction

We use the observation that spurious correlations introduce clusters to relax the requirements of hand-crafted environments for IRM. We define the *minority set* $D_m$ as the union of all *minority cases* over all clusters. Thus, the *minority set* contains all *conflict* samples, so $D_m$ is expected to have an

---

[1]Arjovsky *et al.* [2019] use two environments with an equal amount of instances in each. $p_e = 0.1$ in one and $p_e = 0.2$ in the other, resulting in an overall $p_e = 0.15$ for ERM training.

[2]We choose $k = 8$ for all experiments in this paper.

| Method | Annot. | Test Acc. |
|---|---|---|
| ERM (baseline) | ✗ | $16.9 \pm 0.5$ |
| Oracle (upper bound) | ✗ ✗ | $72.7 \pm 0.3$ |
| ERM ($D_{\text{balance}}$) | ✗ | $65.7 \pm 0.7$ |
| IRM* | ✓ | $66.9 \pm 2.5$ |
| DecAug* | ✓ | $69.6 \pm 2.0$ |
| EIIL* | ✗ | $68.4 \pm 2.7$ |
| IRM ($D_m$, $D_{\text{balance}}$) | ✗ | $68.0 \pm 1.1$ |

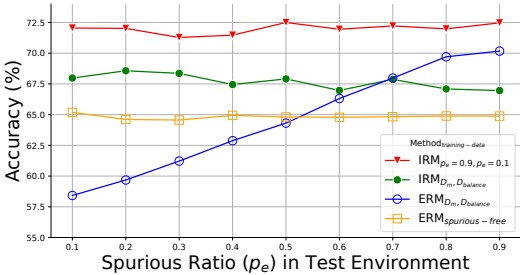

(a) Standard test environment $p_e = 0.9$     (b) Varying $p_e$ in test environments

Figure 2: Predictive performance (average over 10 runs) on **(a)** the standard test set. ✓ indicates that the method requires annotations of the environments, whereas ✗ does not; ✗ ✗ for the Oracle means the method is trained and evaluated on *gray images* with $n_y = 0.25$. We highlight models using our sampling approach. * indicates values from the original paper. **(b)** varying test environments.

inverse correlation between colors and labels compared to the training set. This also means that $D_m$ contains a spurious correlation that is opposite to the one in the original training set. To obtain a more balanced sample, we additionally sample *dominant cases*, i.e., cases that belong to the majority class in this cluster. Overall, we sample as many dominant cases as minority cases. Our balanced set $D_{\text{balance}}$ is the union of minority and dominant cases, and is designed to be likely class-balanced and balanced between *conflict* and *non-conflict* samples. To account for IRM's multiple environments, we train IRM with $D_m$ and $D_{\text{balance}}$.

## 5 Predictive Performance

We follow the training architecture and data partitioning of previous work [Arjovsky *et al.*, 2019], using a simple CNN and the official test set with color correlation $p_e = 0.9$. First, we train the baseline ERM model on the training set with $p_e = 0.15$. We compare our approach to IRM [Arjovsky *et al.*, 2019] and DecAug [Bai *et al.*, 2021], both of which require annotations, and to EIIL [Creager *et al.*, 2021], which infers suitable environments without annotations.

The results in Tab. 2a show that IRM training with our sampling approach $D_m$ and $D_{\text{balance}}$ is competitive to methods that require annotations (slightly better than standard IRM and slightly worse than DecAug) and on par with the annotation-free method EIIL. In contrast to EIIL, our approach neither poses restrictions on the reference model, nor does it require to train additional parameters.

We analyze the effectiveness of our sampling approach on two models: ERM trained with $D_{\text{balance}}$ and IRM trained with the two environments $D_m$ and $D_{\text{balance}}$. The results are shown in Tab. 2a (highlighted rows). Our sampling of an approximately balanced set $D_{\text{balance}}$ improves the accuracy of the ERM model from 17% to 65%. This result is even close to that of invariant learning methods such as IRM. The performance is also comparable to the performance of ERM trained with a *hand-crafted balanced* set ERM$_{\text{spurious-free}}$ as shown in Fig. 2b, indicating that our sampling approach is capable of constructing a *truly* balanced set.

## 6 Analysis and Discussion

**ERM and IRM.** By construction, the training and test distributions of ColoredMNIST have an inverse distribution of colors and labels. The environments constructed by our method may have an advantage, because we choose counter-spurious samples, thereby implicitly forming an inverse distribution. Therefore, we verify the ability to learn invariant features by varying the amount of spurious correlations in the test set. The ultimate goal of invariant learning is to have consistent performance across different environments. Fig. 2b shows the accuracy over test sets with different spurious ratios $p_e$ from 0.1 to 0.9 (official test set). ERM$_{D_m, D_{\text{balance}}}$ and IRM$_{D_m, D_{\text{balance}}}$ are trained with the sampled environments defined in Sec. 4. IRM requires (at least) two different environments for invariant learning ($D_m$, $D_{\text{balance}}$) and ERM is trained with the concatenation of $D_m$ and $D_{\text{balance}}$. ERM$_{\text{spurious-free}}$ is trained with a *hand-crafted balanced* subset, which does not contain spurious correlations. With

the concatenation of $D_m$ and $D_{\text{balance}}$, the spurious correlation learned by $\text{ERM}_{D_m, D_{\text{balance}}}$ is opposite to the correlation in the training set, but aligns with the correlation in the test set (not as strong as in $D_m$ alone though). Accordingly, $\text{ERM}_{D_m, D_{\text{balance}}}$ shows a clear trend following the increase of the spurious ratio, while $\text{ERM}_{\text{spurious-free}}$ and $\text{IRM}_{D_m, D_{\text{balance}}}$ are largely consistent over different test environments and IRM always out-performs $\text{ERM}_{\text{spurious-free}}$. This implies that ERM is sensitive to the spurious correlation (ratio) in the training and test set while IRM with multiple environments is robust against particular correlation ratios in individual environments and generally less dependent on spurious features.

**Environments Optimization.** As Arjovsky *et al.* [2019] discuss, IRM cannot prevent models from learning spurious correlations, even when trained with hand-crafted environments. This is also evidenced by a performance gap between IRM and the Oracle model (cf. Tab. 2a). In Fig. 2b we evaluate another choice of hand-crafted environments, $\text{IRM}_{p_e=0.9, p_e=0.1}$, with a training set comprised of two environments with $p_e = 0.9$ and $p_e = 0.1$ respectively. We observe an increase in performance (Fig. 2b, red graph) with this choice of environments, being even on par with the Oracle (73%). This observation highlights the importance of defining a good combination of environments for invariant learning in general. It sets an interesting future research direction to investigate the explicit condition for environment construction in invariant learning, and more importantly, how to obtain the most beneficial environments without additional annotation requirements.

**Environment Inference without Annotations.** EIIL infers suitable environments by a learnable probability distribution over the assignments of instances to environments $q_i(e') := q(e'|x_i, y_i)$, indicating the probability that the $i$-th sample belongs to environment $e'$. EIIL estimates $q$ by *maximizing* the regularization term in IRMv1 w.r.t a reference model $\tilde{\Phi}$. The goal of this maximization is to infer environments that are governed by (supposedly spurious) features that maximally violate the invariant principle. That is, a (supposedly spurious) feature should be correlated with a particular class in one environment and with a different class in the other. Subsequent IRM training on these environments then allows to extract invariant features. For this process to be effective, the reference model $\tilde{\Phi}$ is required to heavily rely on spurious correlations. In practice, an ERM model in an early training stage (early stopped) is chosen for $\tilde{\Phi}$. However, the availability of such a model strongly depends on the dataset and careful tuning of early-stopped criteria. The reason is that the ERM model initially focuses on spurious features during early training, resulting in significant performance disparities across subgroups, which provides a signal for EIIL to infer effective environments. However, this learning signal weakens when using a well-trained model [Creager *et al.*, 2021].

In contrast, our approach infers environments from representation clustering (cf. Sec.4.1), without posing restrictions on the reference model, and the clustering overhead is small compared to the estimation of $q$.

## 7 Conclusion

In this paper, we introduced a novel strategy to identify conflict samples in the training dataset in the presence of spurious correlations, based on the observation that instances tend to cluster more strongly w.r.t. spurious features than w.r.t. class labels in the learned representation space. Our approach allows for the easy construction of sub-datasets with varying spurious correlation ratios without explicit annotations, forming multiple environments suitable for invariant risk minimization. In future work, we plan to validate whether the method successfully extends to more general and complex scenarios, e.g., multi-class classification tasks, varying strength of spurious correlations, and in the presence of multiple spurious correlations. We further aim to identify the limits of our approach, i.e., boundary cases where the method fails.

## Acknowledgments and Disclosure of Funding

This project was partially supported by Hessian.AI Connectom Fund 2024.

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
