# OpenReview forum: "Invariant Learning with Annotation-free Environments"
_NeurIPS.cc/2024/Workshop/UniReps — UniReps_

### Official Review · Reviewer_XTLA · 2024-09-28
**Interesting idea but with weak experiments**

**Rating:** 5
**Confidence:** 4

**Review:**

The paper introduces an approach to infer environments for invariant learning without the need for explicit annotations, which is better than prior works that rely on pre-defined environments. Here are some issues.
1. The experiments focus on ColoredMNIST, which is simple. It would be beneficial to extend the analysis to more real-world datasets to validate the generalizability of the approach.
2. The method depends on strong spurious correlations, e.g., the reliance on clusters formed by spurious features. But how the method performs in scenarios with weaker or more complex spurious correlations.
3. Including a section analyzing situations where the method fails or performs suboptimally is better.

---

> ### Author Response · Authors · 2024-11-05
>
> This paper is an initial investigation of the method on the simple ColoredMNIST dataset. We added a discussion of why we deem the method to work in more complex scenarios as well to section 4 and plan to investigate those (and also boundary cases, where the method fails) in future work.

---

### Official Review · Reviewer_W6JM · 2024-10-03
**Reducing Reliance on Spurious Correlations**

**Rating:** 7
**Confidence:** 4

**Review:**

**Summary:**

Given a dataset with large spurious correlations, the work proposes a way of constructing a curated dataset,
which reduces the reliance on spurious correlations of a model trained on this new dataset.
The reliance on spurious correlations is compared with the same model trained on the original dataset and with
a model trained with Invariant Risk Minimization (IRM).
The construction of the curated dataset depends on a model trained on the original dataset.


**Questions and Suggestions:**

Q1: You use k-means on some model embeddings to choose your training dataset composition.
Since k-means usually does not work very well in high dimension, have you considered what one should do in the
case of a high-dimensional embedding space? Or do you have a reason why it should still work?

Q2: In section 2, I suggest being more clear on the difference between Empirical Risk Minimization and Invariant Risk Minimization.
For example by explicitly stating the optimization problem for Empirical Risk Minimization.

Q3: I suggest rewriting the first sentence of your abstract:
"Invariant learning across environments is a promising approach to improve domain generalization compared to ERM."
It is not clear what you mean by "environments" here. Also, please use the full name and not an abbreviation the first time you mention a concept,
in this case Empirical Risk Minimization (ERM).

Q4: I suggest rewriting the paragraph beginning with "However, these approaches require explicit..." on line 28, since it is difficult to read in its current form.
Instead of repeating the difficulties ((i) and (ii)), I suggest beginning with your example, e.g.
"A realistic example of spurious features ..." and only clarifying the difficulties after.

Q5: When you write: ERM_{D_m, D_balance}, do you mean that the model was trained on a concatenation of D_m and D_balance?
So ERM_{D_m, D_balance} is actually trained on a dataset which has the inverse spurious correlation of the original dataset,
just less strong than in the original data? This is what it looks like in figure 2 (b). Please be more clear about it.

Q6: How would you extend this method to classification with more than two classes? If you don't know how to extend the method, it should be mentioned as a limitation.


**Strengths:**

S1: The accuracy results presented are averages over 10 runs, which gives a good feel for the amount of variation which can be expected
in this experimental setting.

S2: Once the first model has been trained, the new curated dataset can be constructed automatically. There is no need for hand-picking samples.


**Weaknesses:**

W1: There is no clearly marked conclusion.

W2: There are several places where the writing is unclear, see Q2-Q5.

W3: The experiments are only on a binary classification problem. The result would be stronger,
if it could be shown that the procedure also works in a setting with more classes.


**Justification:**

I recommend this paper to be accepted to the workshop with the added clarifications.

The main weaknesses can be fixed with rewriting.

---

> ### Author Response · Authors · 2024-11-05
>
> **Answers to Questions**
>
> A1: “k-means usually does not work very well in high dimension” is rooted in the curse of dimensionality (most instances have near identical distance, rendering distance-based comparison meaningless). Whether this curse is present in machine learning applications (where data generation processes are not independent per dimension) has been called into question at least (see e.g., [1]). We expect classification models to learn patterns in the training data and, therefore, representations to cluster in dense (enough) regions that allow meaningful distance-based comparison.
>
> A2 - A4: We adjusted the writing according to your suggestions.
>
> A5: Correct. We added a more explicit elaboration to Section 6 ERM and IRM.
>
> A6: The goal of the method is to construct (at least) 2 different environments with different levels of spurious correlations, i.e. in the paper we choose D_m containing all conflict samples and D_balance containing (probably) zero spurious correlation. The goal of IRM is to learn invariant features across different environments, so even in a k-class (k > 2) classification task, having 2 environments is enough to run the method.
>
> We added the conclusion (Section 7)
>
> [1] Lin, W. Y., Liu, S., Ren, C., Cheung, N. M., Li, H., & Matsushita, Y. (2021). Shell theory: A statistical model of reality. IEEE Transactions on Pattern Analysis and Machine Intelligence, 44(10), 6438-6453.

---

### Official Review · Reviewer_qZDn · 2024-10-06
**This submission proposes an invariant learning method in annotation-free environments by first identifying counter-spurious correlations to construct environments that enable learning invariant features.**

**Rating:** 6
**Confidence:** 3

**Review:**

**Strengths:**
- Clear and well-defined problem statement.
- Thorough discussion of potential future work.

**Comments:**
The method section could benefit from improved clarity and structure to enhance understanding and better highlight the contributions.

**Limitations:**
The results remain limited compared to baseline models.

---

> ### Author Response · Authors · 2024-11-05
>
> [Limitations]
>
> While some baselines in previous studies show similar performance, our method offers distinct advantages over other approaches.
> - Compared to DecAug or IRM, our method does not require environment labels or spurious feature annotations.
> - Compared to EIIL, our approach reduces the computational complexity and eliminates the need for a heavily spurious-dependent reference model, as discussed in Section 6 (Environment Inference without Annotations).
>
> We highlighted our key contribution in the method section of the camera-ready version.

---

### Author Response · Authors · 2024-11-05

We thank all reviewers for their insightful comments and have provided additional details and incorporated changes in individual responses.

---

### Decision · Program_Chairs · 2024-10-10

**Decision:**

Accept

**Comment:**

In light of the positive reviewers' feedback and relevancy of the submission, we are pleased to accept this paper for presentation at UniReps 2024. We kindly ask the authors to incorporate the reviewers' suggestions and feedback in the final camera-ready version of the manuscript.